# Local Complexity Estimation Based Filtering Method in Wavelet Domain for Magnetic Resonance Imaging Denoising

**DOI:** 10.3390/e21040401

**Published:** 2019-04-16

**Authors:** Izlian Y. Orea-Flores, Francisco J. Gallegos-Funes, Alfonso Arellano-Reynoso

**Affiliations:** 1Escuela Superior de Ingeniería Mecánica y Eléctrica, Instituto Politécnico Nacional Av. IPN s/n, Edificio Z, acceso 3, 3^er^ piso; SEPI-Electrónica, Col. Lindavista, 07738 Ciudad de México, Mexico; 2Instituto Nacional de Neurología y Neurocirugía, Av. Insurgentes Sur 3877, Col. La Farma, 14269 Ciudad de México, Mexico

**Keywords:** local complexity estimation, wavelet, MRI denoising

## Abstract

In this paper, we propose the local complexity estimation based filtering method in wavelet domain for MRI (magnetic resonance imaging) denoising. A threshold selection methodology is proposed in which the edge and detail preservation properties for each pixel are determined by the local complexity of the input image. In the proposed filtering method, the current wavelet kernel is compared with a threshold to identify the signal- or noise-dominant pixels in a scale providing a good visual quality avoiding blurred and over smoothened processed images. We present a comparative performance analysis with different wavelets to find the optimal wavelet for MRI denoising. Numerical experiments and visual results in simulated MR images degraded with Rician noise demonstrate that the proposed algorithm consistently outperforms other denoising methods by balancing the tradeoff between noise suppression and fine detail preservation. The proposed algorithm can enhance the contrast between regions allowing the delineation of the regions of interest between different textures or tissues in the processed images. The proposed approach produces a satisfactory result in the case of real MRI denoising by balancing the detail preservation and noise removal, by enhancing the contrast between the regions of the image. Additionally, the proposed algorithm is compared with other approaches in the case of Additive White Gaussian Noise (AWGN) using standard images to demonstrate that the proposed approach does not need to be adapted specifically to Rician or AWGN noise; it is an advantage of the proposed approach in comparison with other methods. Finally, the proposed scheme is simple, efficient and feasible for MRI denoising.

## 1. Introduction

Magnetic resonance imaging (MRI) is a powerful medical imaging modality used to produce detailed images of soft tissues and anatomical body structures that can be visualized non-invasively at the millimeter scale [1,2]. MRI processing provides detailed quantitative brain analysis for accurate disease diagnosis [3,4] (i.e., brain tumor diagnosis [5], Alzheimer’s disease (AD), Parkinson’s disease, multiple sclerosis [6], dementia, schizophrenia, brain disorder identification and whole brain analysis of traumatic injury), detection, treatment planning and classification of abnormalities (i.e., extracting tissues like white matter (WM), gray matter (GM) and cerebrospinal fluid (CSF)) [3].

In clinical evaluation and neuroscience research, MRI images are often corrupted by several artifact sources, such as intensity inhomogeneity, abnormal tissues with heterogeneous signal intensities, non-ideal hardware characteristics and the poor choice of scanning parameters [2,3,7]. In order to improve the quality of noisy MRI images to facilitate clinical diagnosis, the MRI pre-processing operations are introduced to improve the qualities of other MRI applications such as segmentation [8], detection [9] and classification [2,10].

Image denoising is a standard pre-processing task for MRI to precisely delineate regions of interest between different brain tissues, to enhance the contrast between regions and to reduce noise, while preserving, as much as possible, the image features as well as structural details [2,10]. 

Many denoising methods for MRI have been proposed in the literature, these methods can be divided into three major classes [11,12]: (1) *filtering techniques* include linear filters (i.e., spatial and temporal methods) and non-linear filters (i.e., anisotropic diffusion filtering (ADF) -based methods) [10], 4th order partial differential equation (PDE) –based methods, non-local means (NLM) –based methods [13] and combination of domain and range filters (i.e., bilateral and trilateral filters); (2) *transform domain methods,* this class consider the curvelet and the contourlet transforms [14,15] and the wavelet transform based methods (i.e., wavelet thresholding, wavelet domain filter, wavelet packet analysis, adaptive multiscale product thresholding, multiwavelet and undecimated wavelet) [7,12,16]; (3) *Statistical methods* such as maximum likelihood estimation approach [17], Bayesian approach [18], linear minimum mean square error estimation approach, phase error estimation approach, nonparametric neighborhood statistics/estimation approach and singularity function analysis [11,18,19]. Additionally, there exist some hybrid methodologies that belong to both NLM-based methods and Statistical approaches [20,21].

The spectrum of applications in medicine and biology of the wavelet transform has been extremely large, it includes the analysis of the electrocardiogram (ECG) and imaging modalities such as positron emission tomography (PET) and MRI [22]. The main difficulty in dealing with biomedical objects is the variability of the signals and the necessity to operate on a case by case basis [22]. On the other hand, the wavelet decomposition is determined by one mother wavelet function and its dilation and shift versions [23]. There are a lot of wavelet families published in the literature, but researchers commonly have difficulty selecting an optimal wavelet for a specific image processing application [23]. The choice of the optimal wavelet function depends on different criteria in several applications and in some of the distinctive properties (i.e., region of support and the number of vanishing moments) of the wavelet function [23,24].

In this paper, we propose the local complexity estimation based filtering method in wavelet domain for MRI denoising. A threshold selection methodology is proposed in which the edge and detail preservation properties for each pixel are determined by the local complexity of the input image. Statistics of standard deviation select the pixels whose values can be changed since low-energy wavelet coefficients correspond to the smooth regions and high-energy wavelet coefficients are in agreement with the signal features of sharp variation (i.e., edges and textures). In the proposed filtering method, the current wavelet kernel is compared with a threshold to identify the signal- or noise-dominant pixels in a scale providing a good visual quality avoiding blurred and over smoothened processed images. We present a comparative performance analysis with different wavelets to find the optimal wavelet for MRI denoising. The purpose of this research is to eliminate the noise in the MR image as much as possible without losing the details corresponding to image features as the structural details, which will be of highly useful in the quantitative brain analysis for accurate disease diagnosis. Numerical experiments and visual results in simulated MR images degraded with different percentages of Rician noise demonstrate that the proposed algorithm consistently outperforms other denoising methods by balancing the tradeoff between noise suppression and fine detail preservation. The proposed algorithm can enhance the contrast between regions allowing to delineate the regions of interest between different textures or tissues in the processed images. The proposed approach shows a satisfactory result in the case of real MRI denoising by balancing the detail preservation and noise removal, with enhancing the contrast between the regions of the image; otherwise, the comparative methods produce smooth results or limited denoising effectiveness. Additionally, the proposed algorithm is compared with other approaches using standard images degraded with different standard deviations of Additive White Gaussian Noise (AWGN) to demonstrate that the proposed approach does not need to be adapted specifically to Rician or AWGN noise, it is an advantage of the proposed approach in the denoising task of both AWGN and Rician noises against other methods. Finally, the proposed scheme is simple, efficient and feasible for the MRI denoising, the obtained results suggest that the application of the proposed method can benefit many quantitative techniques (i.e., segmentation, tractography or relaxometry) that can take advantage from the denoising and enhanced data produced for the application of the proposed method. 

The paper is organized as follows. Section 2 designs the proposed filtering algorithm to MRI denoising. Section 3 presents the performance results in image filtering. Finally, Section 4 concludes the paper.

## 2. Proposed Method

Discrete wavelet transform (DWT) is an implementation of the wavelet transform using a discrete set of wavelet scales and translations [7]. DWT decomposes an image in different (approximation and detail) sub-bands at different frequencies (scales) with the help of high pass and low pass filters [25]. Figure 1a presents the DWT scheme using high pass filters to extract the high frequency information (i.e., edges and fine details of the image) and low pass filters to obtain the low frequency information (i.e., the low pass representation or the approximation of the image), these filters are first applied in one dimension and then in another one [25]; and Figure 1b depicts the decomposition of a noisy image using the DWT in four wavelet sub-bands labeled as the low-low (LL*_s_*) sub-band correspond to the approximation sub-band and the low-high (LH*_s_*), high-low (HL*_s_*) and high-high (HH*_s_*) sub-bands correspond to horizontal, vertical and diagonal details of the image, respectively, s={1,2,…,S} is the scale and *S* represents the coarsest scale [12,25]. 

In the DWT implementation, a standard decimated filterbank algorithm is used (see Figure 1a) [22], a high pass filter *g*[*n*] and a low pass filter *h*[*n*] are applied to a noisy signal *y*[*n*] in the following way [26]
(1)$$y_{high}\left[k\right]=\sum_{n}y\left[n\right]\;g\left[2k-n\right]\;y_{low}\left[k\right]=\sum_{n}y\left[n\right]\;h\left[2k-n\right]$$
where $y_{high}\left[k\right]$ and $y_{low}\left[k\right]$ are the outputs of the high pass and low pass filters, respectively.

In the wavelet thresholding methods, the detail coefficients are processed with soft or hard thresholding to estimate the signal components [7]. The DWT denoising procedure depends upon the usage of wavelet function and thresholding [12]. The wavelet functions are used for estimating the noiseless coefficients from noisy wavelet coefficients in wavelet domain. Various threshold selection methodologies have been proposed to minimize the contribution of noise such as VisuShrink, SureShrink, BayesShrink and NeighShrink [7,12].

We propose a threshold selection methodology in which the edge and detail preservation properties for each pixel are determined by the local complexity of the input image. In the proposed method the current wavelet kernel is compared with a threshold to identify the signal- or noise-dominant pixels in a scale providing a good visual quality avoiding blurred and over smoothened processed images. The steps of the proposed algorithm are given as follows.

*Step 1. Apply the DWT*. Let obtain the decomposition of the noisy image using the DWT and choose the sub-band HH_1_ to realize the next steps.

*Step 2. Compute the standard deviation*. The standard deviation of wavelet coefficients shows the corresponding energy of wavelet coefficients (i.e., low-energy wavelet coefficients appertain to the smooth regions and high-energy wavelet coefficients appertain to the edges and textures). Let compute the standard deviation $\sigma_{p}$ in the sub-band HH_1_ where p is the current kernel. The standard deviation $\sigma_{p}$ is computed using a 3 × 3 kernel according with Figure 2
(2)$$\sigma_{p}=\sqrt{\sum_{m_{p}=1}^{n_{p}}{\left(y_{m_{p}}-\bar{y}\right)}^{2}/n_{p}}$$
where $y_{m_{p}}$ is the *m_p_*-th element of the current kernel p={1,2,…,N}, N is the total number of kernels in the sub-band HH_1_, $\bar{y}=\sum_{m_{p}=1}^{n_{p}}y_{m_{p}}/n_{p}$ is the mean value of the current kernel and $n_{p}=9$ is the number of elements in the kernel. 

*Step 3. Compute the threshold.* The pixels are classified using a threshold based on the local values of the standard deviations of all kernels in the sub-band HH_1_. The median value of the standard deviations has been chosen for this purpose. The median is used as robust estimation of the energy of the wavelet kernel coefficients given by its local standard deviation [27,28]. The threshold T selects the pixels whose values are considered as noisy
(3)$$T=\mathrm{MED}\left\{\sigma_{1},\sigma_{2},\sigma_{3},\ldots ,\sigma_{N}\right\}$$
where MED is the median.

*Step 4. Apply condition to the current kernel*. The proposed condition provides good noise removal, while the edges and the fine details are preserved. The proposed condition to provide denoising is given as follows,
(4)$$w_{co}\;=\{\begin{array}{cc} 0, & \alpha_{p}>T\\ w_{c}, & otherwise \end{array}$$
where $w_{co}$ is the output of proposed procedure, $w_{c}$ is the original wavelet coefficient, $\alpha_{p}=\left|\sigma_{pc}-\mathrm{MED}\left\{\sigma_{p1},\sigma_{p2},\ldots ,\sigma_{pn}\right\}\right|$ is the proposed noise estimation parameter, it can be used as an impulsive noise detector when the impulsive noise levels are high [28], it verifies the difference between the value of the median of coefficients and the central coefficient in terms of standard deviation values; $\sigma_{pc}$ is the standard deviation located in the center of current kernel p, this is, each kernel provides such estimation; and $\left\{\sigma_{p1},\sigma_{p2},\ldots ,\sigma_{pn}\right\}$ are the standard deviations contained in the current kernel p.

We note that the high-energy wavelet coefficients in the sub-band HH_1_ involve noise and the edges and textures. The proposed condition (4) distinguishes when a wavelet coefficient (pixel) is noisy or is a detail (edge or texture) in the following way: If the value of the proposed noise estimation parameter $\alpha_{p}$ is bigger than the threshold T, then the current kernel p in the sub-band HH_1_ is classified as noisy and in such positions the values of the wavelet coefficients are setting in zero (see Figure 2). Otherwise, the wavelet coefficients of this kernel are classified as details and these are unaltered.

*Step 5. Compute the Inverse Discrete Wavelet Transform (IDWT).* We obtain the restored image applying the IDWT according to the wavelet decomposition.

## 3. Simulation Results

The proposed local complexity estimation based filtering method in wavelet domain is compared with some reference approaches commonly used in the literature in terms of objective performances given by PSNR (Peak Signal-to-Noise Ratio) and SSIM (Structural Similarity Index) [29] and subjective visual denoising results. The methods used to compare our approach were computed and used in accord with their references. Also, the parameters required by each comparative algorithm are set equal to the values assumed in such references. The reason for choosing these methods to compare them with the proposed one is that their performances have been compared with various known methods and their advantages have been demonstrated.

Four tests have been proposed to determine the performance of the proposed approach. *First*, a comparative performance analysis in a MRI database is done using the DWT with different wavelets to find the optimal wavelet for MRI denoising; *Second*, the proposed algorithm is compared with other approaches using standard images degraded with different standard deviations of Additive White Gaussian Noise (AWGN); *Third*, comparative results in simulated MR images degraded with Rician noise are obtained to evaluate our proposal; and *Fourth*, a real case of MRI denoising is shown to demonstrate the capabilities of noise filtering of the proposed approach against other methods.

We note that the use of AWGN with different standard deviations is proposed to demonstrate the robustness of the proposed approach in the denoising of standard images in comparison with other methods published recently. In the case of simulated MRI, the Rician noise is built from white Gaussian noise in the complex domain. The proposed approach does not need to be adapted specifically to Rician noise, it is an advantage of the proposed approach in the denoising task of AWGN and Rician noises against other methods. For this reason, we implement these test to determine the performance of the proposed approach.

### 3.1. Comparative Performance of Different Wavelets

In order to analyze different wavelets for MRI denoising, we utilize a database provided by the National Institute of Neurology and Neurosurgery of Mexico [30]. The real dataset has been recorded by using a Philips Achieva MRI 1.5T scanner with the following parameters: Echo and Repetition Times equal to 102 and 5000 ms, respectively, the Field Of View is 276 × 270 mm and image size of 512 × 512 pixels. This dataset has 900 MRI of three patients (300 MRI for each patient) in a DICOM (Digital Imaging and Communications in Medicine) format. We evaluate the wavelets Haar, Daubechis 2 (DB2), Daubechis 4 (DB4), Symlets 2 (SYM2), Symlets 4 (SYM4), Coiflets 1 (COIF1) y Coiflets 2 (COIF2) in the DWT to realize the MRI denoising. During the wavelet decomposition process, the detail coefficients can be processed with soft or hard thresholding to estimate the signal components for effective denoising [7,12]. Our aim is to find the optimal wavelet according to the best PSNR and SSIM values for a hard threshold of T=0, and with this, all high frequency information (the noise, edges and fine details of the image) of the horizontal (LH_1_), vertical (HL_1_), and/or diagonal (HH_1_) details is eliminated. After numerous simulations, we decide to apply this procedure in the wavelet coefficients of HH_1_ sub-band, the value of T=0 was chosen only to find the optimal wavelet. With this wavelet, in Section 3.2 we apply the proposed threshold selection methodology to preserve the edges and fine details of the image.

Table 1 presents the average performance results in MRI denoising on the MRI database in terms of PSNR (Peak Signal-to-Noise Ratio) and SSIM (Structural Similarity Index). From Table 1, one can see that the best average PSNR and SSIM performances are given for DB4 and Haar wavelets, respectively. The differences between the results obtained using the two objective quality measures are given because the PSNR is sensitive to the energy of errors instead of real information loss in spite of it is still employed “universal” regardless of its questionable performance in several image applications and SSIM is designed to model any image distortion as a combination of the loss of correlation, luminance distortion and contrast distortion factors, it is applicable to different image processing applications because it does not depend on the images being tested, the viewing conditions or the individual observers [29]. For these reasons, there are no coincidences between both quality measures. 

Figure 3 depicts the visual results applying the DWT with different wavelets in a MRI image in terms of PSNR and SSIM. The visual results reveal that the best performances of noise suppression and image distortion are given for DB4 and Haar wavelets, respectively. These results are in concordance with the PSNR and SSIM performances of Table 1.

### 3.2. Comparative Performance in Standard Images

To evaluate the proposed algorithm in the task of AWGN denoising, we apply the test presented in Reference [12] considered the same data and conditions. For this purpose, we use ten standard images (Lena, Jetplane, Mandrill, House, Boat, Lake, Peppers, Barbara, Pirate and Texture) of size 512 × 512 pixels degraded with the standard deviation σ={10,20,30,40,50} of AWGN with zero mean. These images present natural noise, artifacts (noise, intensity, color inhomogeneity in the regions, regions with similar textures, shadows, object reflections, etc.) and diverse content such as fine structures (parallel edges), homogenous areas, texture details and structural information [12]. Comparative performance analysis is carried out for a) wavelet-based approaches such as, VisuShrink with hard threshold [31], BayesShrink [32] and NeighSureShrink [33] and b) NLM (nonlocal means) -based approaches, such as, the standard NLM [34], NLM-DCT (NLM-Discrete Cosine Transform) [35] and NLM-DCT-WEIGHTED (NLM-DCT-Weighted) [12].

Table 2 shows PSNR and SSIM performances for the proposed method with the use of different wavelets in the standard images degraded with a different standard deviation of AWGN. From Table 2, we observe that the best PSNR performance is for the proposed method with DB4 wavelet and in the case of SSIM performance is in favor of the DB4 wavelet in the most of cases (σ={10,20,30}) followed by DB2 and Haar wavelets for σ={40,50}.

We note that Table 1 and Table 2 show PSNR and SSIM performances but the first one presents the performances on MRI images using the DWT with a hard threshold of T=0 to find the best wavelet and the second one shows the performance results for the proposed denoising method on standard images. From these results, we can conclude that with the best (Haar and DB4) wavelets can denoise images of different kind degraded with Rician noise (MRI images) and AWGN (standard images). Section 3.3 will confirm the findings of Table 1 but using the proposed denoising method instead of the hard threshold.

Figure 4 presents the PSNR and SSIM performance analysis for the proposed method and other ones used as comparative in the ten standard images degraded with σ={10,20,30,40,50} of AWGN. We show experimental results in the images Lena and House, we observe in Figure 4a,b that in the case of image Lena the best PSNR performance is for the proposed method with DB4 wavelet and for the SSIM performance the proposed method outperforms other methods in the case of σ={20,30,40} of AWGN; and Figure 4c,d shows that the best PSNR and SSIM performances are for the proposed method with DB4 wavelet in the image House for all standard deviations of AWGN. Then, we provide the average PSNR and SSIM performances for each standard deviation of AWGN using the ten standard images, these results are given in Figure 4e,f where the proposed method with DB4 wavelet provides the best results in terms of PSNR and SSIM performances for each noise level followed by the proposal with Haar wavelet. Figure 4g,h presents the average, minimum and maximum PSNR and SSIM values computed for each denoising method using the ten images. Finally, the results reveal that the proposed method outperforms other denoising methods used as comparative, in the case of average PSNR is in favor of proposed method from 1.06 to 2.4 dB in comparison with the best comparison method (NLM-DCT-Weighted) for the five levels of AWGN and the average SSIM changes from 0.007 to 0.042 in favor of proposed method in comparison with NLM-DCT-Weighted.

Figure 5 depicts the visual results obtained with different denoising algorithms in the images Lena, Mandrill, Lake, Pirate and Texture, degraded with noise level *σ* = 20 according to Figure 4. The denoised images obtained with the proposed method (DB4 wavelet) have better visual qualities in terms of denoising and fine detail preservation in comparison with other algorithms used as comparative. Moreover, the proposed methodology has the best capability for preserving edges and fine structural details and it enhances the contrast between regions of different texture. It is due to the localization property of wavelets and the proposed condition used to classify the pixels as noisy or details. 

### 3.3. Comparative Performance in Simulated MRI

In this subsection, we realize two tests using simulated MR images from the BrainWeb database [36] and we compare our proposal with different state-of-art denoising methods using different percentages levels of Rician noise.

*Test* 1: We implement the test realized in Reference [37] with the same data and under the same conditions. In this case, we compare the proposed method with the standard NLM [34], UNLM (Unbiased NLM) [38] and UNLMDCT (UNLM Discrete Cosine Transform) [37] denoising algorithms using three images of 217 *×* 181 pixels simulated from the BrainWeb database [36] and degraded with 3%, 6%, 9%, 12%, 15% and 18% of Rician noise: (a) T1-weighted MR image, (b) T2-weighted MR image and (c) proton density-weighted (PD-weighted) MR image. Figure 6 shows the PSNR performance of various denoising methods in the simulated MR images, these results reveal that the proposed method with DB4 wavelet provides better PSNR performance for all percentages levels of Rician noise in comparison with other methods used as comparative, this is, the PSNR changes in favor of the proposed method from 0.16 to 1.97 dB in comparison with the best comparison method (UNLMDCT) for the six levels of Rician noise in the three tested images. Figure 7 depicts the visual results in the case of the T1-weighted and PD-weighted MR images degraded with 6% of Rician noise. This Figure shows that in the case of the T1-weighted MR image, the denoised image with the proposed method provides better noise removal and fine detail preservation and allowing the enhancement between different regions corresponding to different tissues in comparison with other algorithms used as comparative. For the PD-weighted MR image, the visual results reveal that the best performance is provided by the proposed method.

*Test* 2: This test is realized according to Reference [39] with the same data and conditions. For this purpose, the proposed method is compared with the ADF (Anisotropic Diffusion Filter) [40], WIENER Filter [41], TV (Total Variation minimization) [42], standard NLM [34] and NLNS (Nonlocal Neutrosophic Set) [39] denoising algorithms using three images of 217 *×* 181 pixels simulated from the BrainWeb database [36]: (a) T1-weighted MR image degraded by 7% of Rician noise, (b) T2-weighted MR image degraded by 9% of Rician noise and (c) T1-weighted MR image with multiple sclerosis (MS) lesion degraded by 15% of Rician noise. Figure 8 presents the PSNR and SSIM performance for the three MR images, the PSNR results indicate that the best performance is a favor of the proposed method, this is, the PSNR changes in favor of proposed method from 1.89 to 2.39 dB in comparison with the best comparison method (NLNS) but the SSIM performance of proposal disappoint in comparison with the NLNS from 0.0435 to 0.1192. The SSIM behavior differs from the PSNR because the PSNR is an objective criterion measurement, whereas the SSIM better captures human perception. Figure 9 depicts the visual results for the three MR images according to the results presented in Figure 8. From Figure 9, one can see that the use of the proposed methodology appears to have better visual qualities in comparison with other algorithms used as comparative.

### 3.4. Comparative Performance in Real MRI

Here, a real case of MRI denoising is presented using the dataset provided in Reference [18]. In this work, Baselice et al. reported comparative results in the real MR image shown in Figure 10a. Denoising visual image results are depicted in Figure 10b–f for the proposed method (DB4 wavelet) and the LMMSE (Linear Minimum Mean Squared Error) [43], BM3D (Block-Matching and 3D) [44], MAP (Maximum A Posteriori estimator) [18] and ADF (Anisotropic Diffusion Filter) [45] denoising algorithms, respectively. From this Figure, the denoising image provided by the proposed approach shows a satisfactory result by balancing the detail preservation and noise removal, by enhancing the contrast between the regions of the image. Otherwise, comparative methods produce smooth results or limited denoising effectiveness. Finally, the obtained results of the proposed approach suggest that it can use as pre-processing stage in MRI applications such as segmentation, detection, and/or classification that can take advantage from the denoising and enhanced data produced for the application of the proposed method.

## 4. Conclusions

We propose the local complexity estimation based filtering method in wavelet domain for MRI denoising. In the proposed methodology, the edge and detail preservation properties for each pixel are determined by the local complexity of the input image to identify the signal- or noise-dominant pixels in a scale providing a good visual quality avoiding blurred and over smoothened processed images. Numerical experiments and visual results in simulated MR images degraded with different percentages of Rician noise have demonstrated that the proposed denoising algorithm provides better image denoising while preserving image features as well as structural details in comparison with other denoising methods proposed in the literature in most cases. This is due to the proposed condition used to classify the pixels as either noisy or details. In the case of real MRI denoising, the proposed approach produces a satisfactory result by balancing detail preservation and noise removal with enhancing the contrast between the regions of the image; otherwise, the comparative methods produce smooth results or limited denoising effectiveness. Additionally, performance results in standard images degraded with different standard deviations of AWGN indicate that the proposed approach does not need to be adapted specifically to Rician or AWGN noise; it is an advantage of the proposed approach in the denoising task of both AWGN and Rician noises, compared with other methods. The main advantages of the proposed scheme for the MRI denoising and other kinds of images are: a) it is simple because in each iteration to decide if the current pixel is noisy or is a detail only compute one standard deviation and two median values, for this reason, we assume that the time complexity of the proposed approach is much less than other methods such as the NLM-based methods; b) it is efficient because the objective results in terms of PSNR and SSIM criteria and subjective results produced by the visual denoised images reveal that the proposed method provides better results in comparison with other methods; and c) it is feasible because the obtained results suggest that the application of the proposed method can benefit many quantitative techniques (i.e., segmentation, tractography or relaxometry) that gain an advantage from the denoising and enhanced data produced for the application of the proposed method.

## Figures and Tables

**Figure 1 entropy-21-00401-f001:**
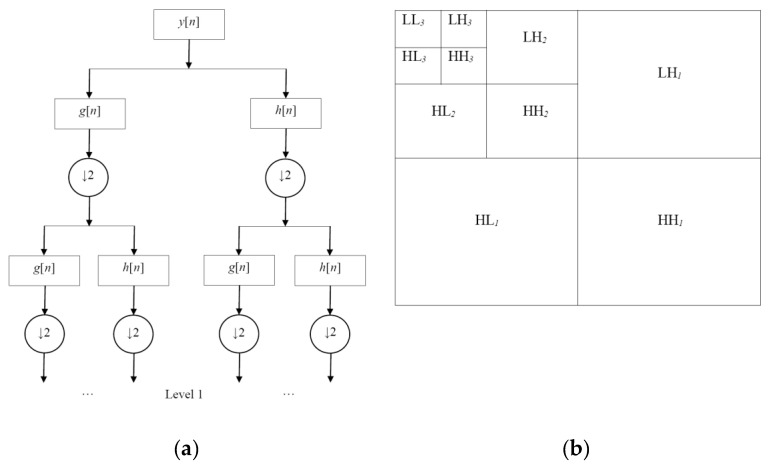
Discrete wavelet transform (DWT): (**a**) DWT scheme using high pass and low pass filters and (**b**) Decomposition of a noisy image using the DWT in four wavelet sub-bands.

**Figure 2 entropy-21-00401-f002:**
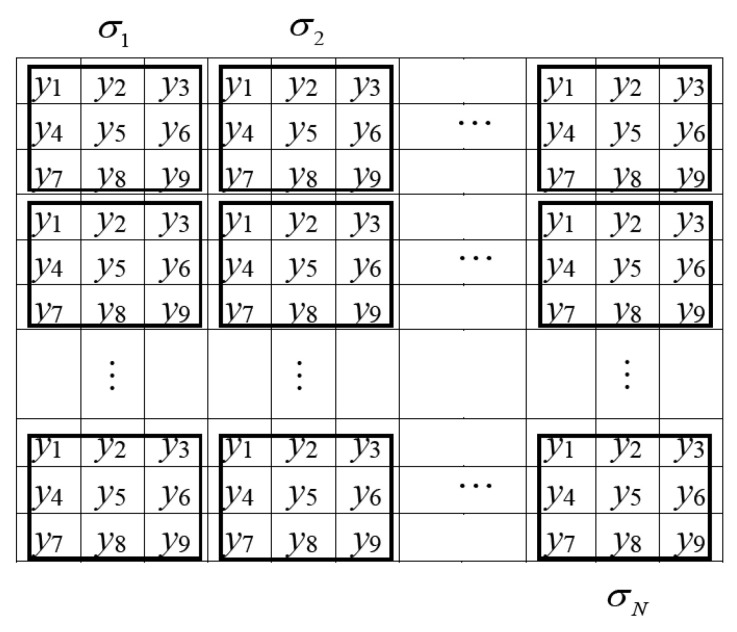
Proposed scheme to compute the standard deviation $\sigma_{p}$ in each kernel p={1,2,…,N} of the wavelet coefficients from the noisy color image.

**Figure 3 entropy-21-00401-f003:**
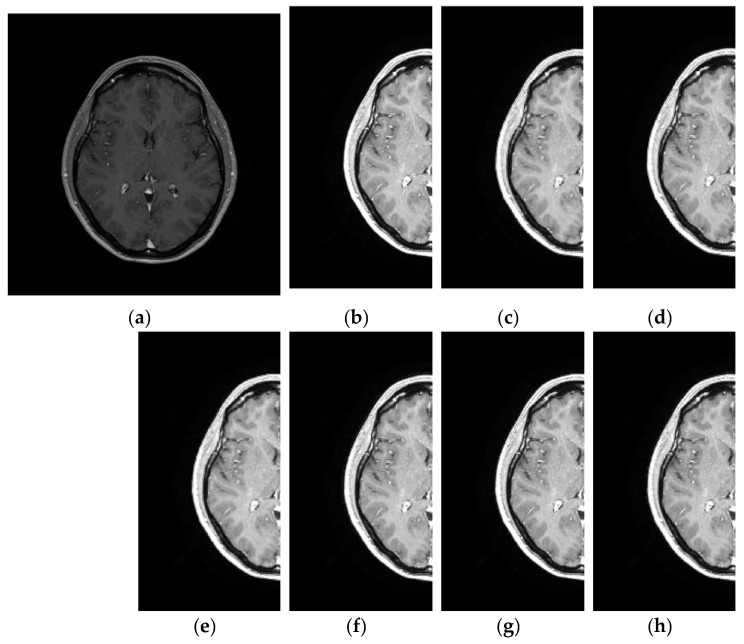
Visual results on magnetic resonance image (MRI) image applying the discrete wavelet transform (DWT) with different wavelets: (**a**) Original MRI, (**b**) Haar (PSNR = 34.812, SSIM = **0.941**), (**c**) DB2 (PSNR = 36.571, SSIM = 0.922), (**d**) DB4 (PSNR = **38.053**, SSIM = 0.913), (**e**) SYM2 (PSNR = 34.598, SSIM = 0.862), (**f**) SYM4 (PSNR = 33.826, SSIM = 0.853), (**g**) COIF1 (PSNR = 34.766, SSIM = 0.852), (**h**) COIF2 (PSNR = 37.054, SSIM = 0.848). The best results are given in bold format.

**Figure 4 entropy-21-00401-f004:**
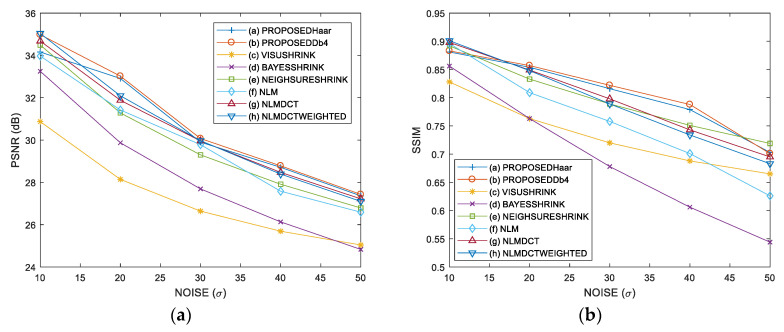
PSNR and SSIM performance analysis of various denoising methods in ten standard images with σ={10,20,30,40,50} of AWGN: (**a**) PSNR performance in the image Lena, (**b**) SSIM performance in the image Lena, (**c**) PSNR performance in the image House, (**d**) SSIM performance in the image House, (**e**) Average PSNR performance using ten images, (**f**) Average SSIM performance using ten images, (**g**) Average, minimum and maximum PSNR values for each denoising method using ten images, (**h**) Average, minimum and maximum SSIM values for each denoising method using ten images.

**Figure 5 entropy-21-00401-f005:**
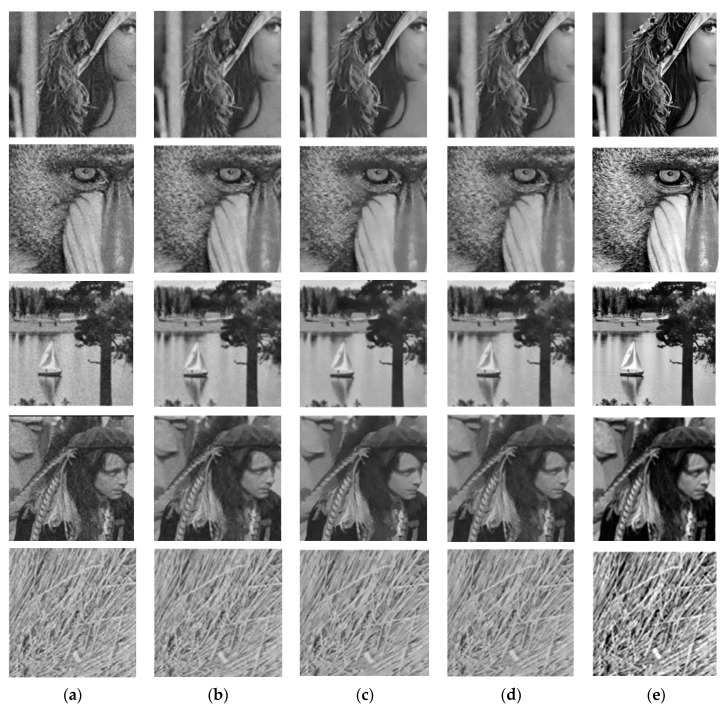
Visual results for different denoising methods in the images Lena, Mandrill, Lake, Pirate and Texture: (**a**) Noisy images with noise level *σ* = 20, (**b**) Denoised images obtained with NeighSureShrink, (**c**) Denoised images obtained with NLM-DCT, (**d**) Denoised images obtained with NLM-DCT-WEIGHTED and (**e**) Denoised images obtained with proposed method (DB4 wavelet).

**Figure 6 entropy-21-00401-f006:**
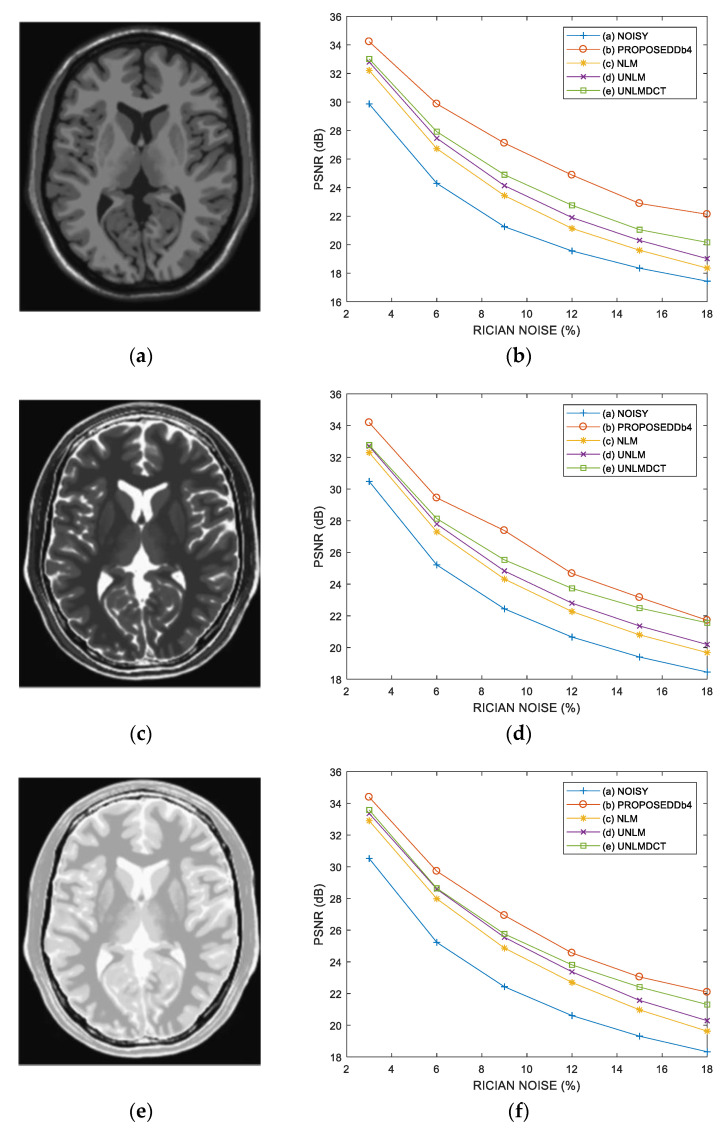
PSNR performance analysis of various denoising methods in simulated MR images degraded with 3%, 6%, 9%, 12%, 15% and 18% of Rician noise: (**a**) Original T1-weighted MR image, (**b**) PSNR performance in the T1-weighted MR image, (**c**) Original T2-weighted MR image, (**d**) PSNR performance in the T2-weighted MR image, (**e**) Original PD-weighted MR image, (**f**) PSNR performance in the PD-weighted MR image.

**Figure 7 entropy-21-00401-f007:**
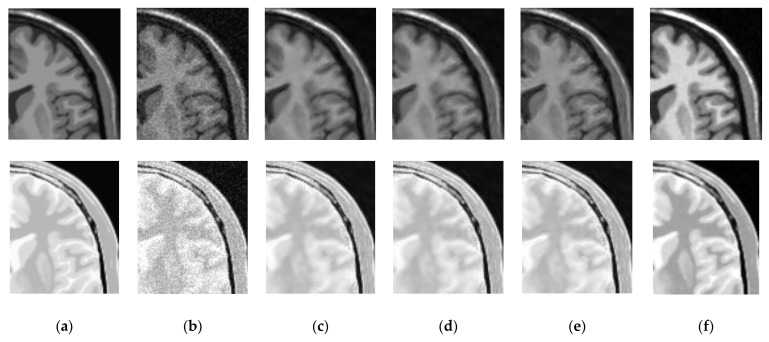
Visual results for different denoising methods in the simulated MR images: (**a**) Original T1-weighted and PD-weighted MR images, (**b**) Noisy MR images degraded with 6% of Rician noise, (**c**) Denoised MR images obtained with NLM, (**d**) Denoised MR images obtained with UNLM, (**e**) Denoised MR images obtained with UNLMDCT and (**f**) Denoised MR images obtained with proposed method (DB4 wavelet).

**Figure 8 entropy-21-00401-f008:**
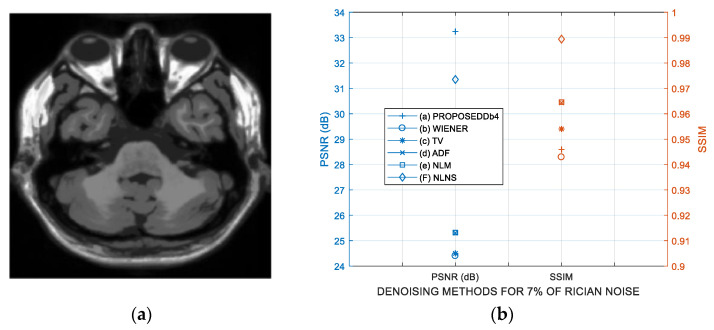
Performance analysis of various denoising methods in simulated MR images degraded with Rician noise: (**a**) Original T1-weighted MR image, (**b**) PSNR and SSIM performances in the T1-weighted MR image degraded by 7% of Rician noise, (**c**) Original T2-weighted MR image, (**d**) PSNR and SSIM performances in the T2-weighted MR image degraded by 9% of Rician noise, (**e**) Original T1-weighted MR image with MS lesion, (**f**) PSNR and SSIM performances in the T1-weighted MR image with MS lesion degraded by 15% of Rician noise.

**Figure 9 entropy-21-00401-f009:**
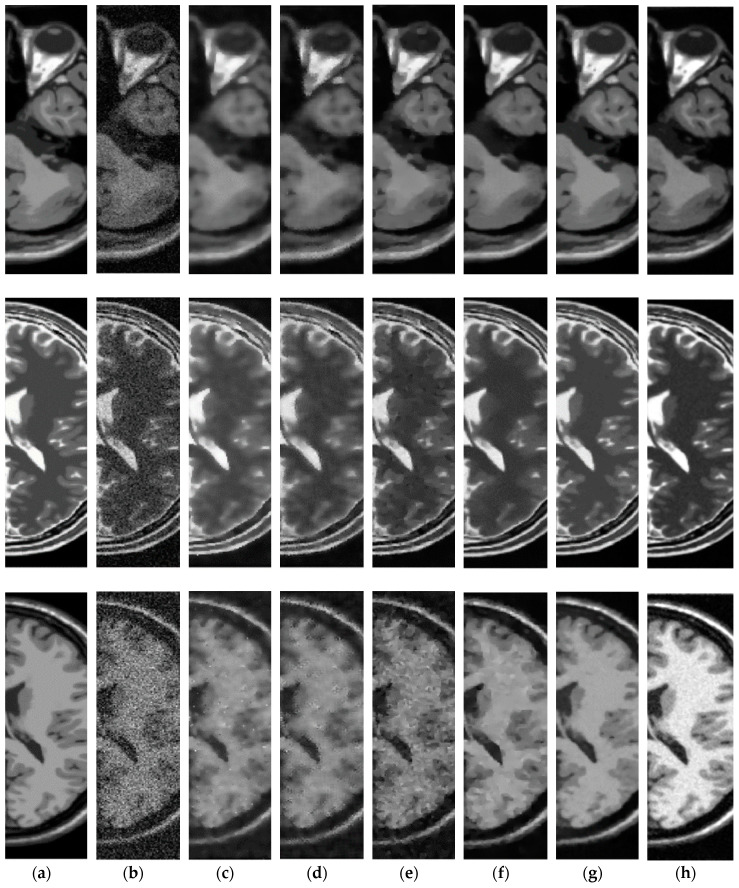
Visual results for different denoising methods in the simulated MR images degraded with different Rician noise: (**a**) Original MR images, (**b**) Noisy MR images, (**c**) Denoised MR images obtained with ADF, (**d**) Denoised MR images obtained with WIENER, (**e**) Denoised MR images obtained with TV, (**f**) Denoised MR images obtained with NLM, (**g**) Denoised MR images obtained with NLNS and h) Denoised MR images obtained with proposed method (DB4 wavelet).

**Figure 10 entropy-21-00401-f010:**
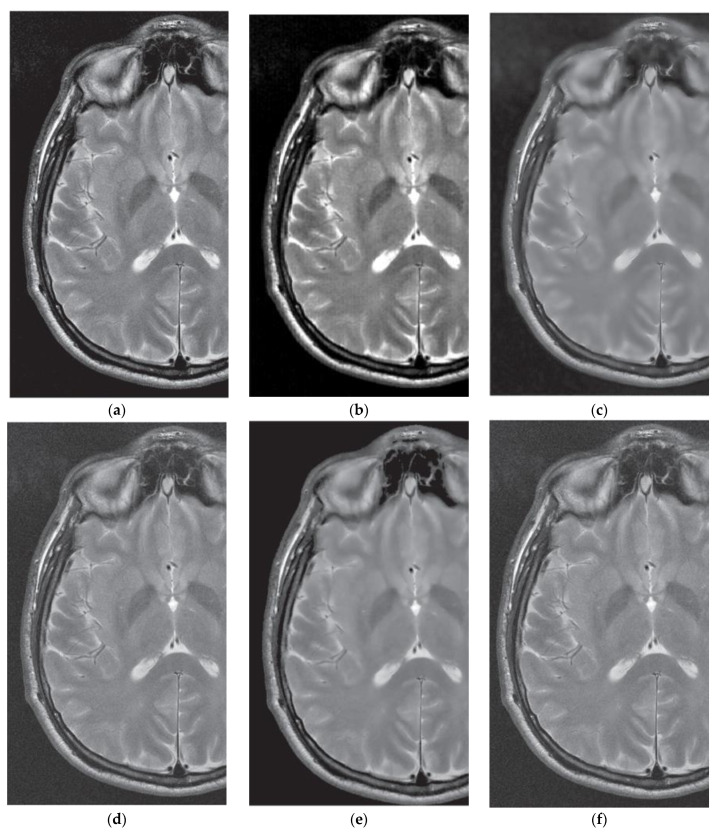
Visual results for different denoising methods in a real MR image: (**a**) Original MR images, (**b**) Denoised MR image obtained with proposed method (DB4 wavelet), (**c**) Denoised MR image obtained with LMMSE, (**d**) Denoised MR image obtained with BM3D, (**e**) Denoised MR image obtained with MAP, (**f**) Denoised MR image obtained with ADF.

**Table 1 entropy-21-00401-t001:** Average peak signal-to-noise ratio (PSNR) and structural similarity index (SSIM) performances obtained from different wavelets. The best results are given in bold format.

Wavelet	PSNR	SSIM
Haar	31.60	**0.941**
DB2	33.50	0.928
DB4	**35.80**	0.936
SYM2	34.50	0.889
SYM4	32.65	0.891
COIF1	31.55	0.878
COIF2	34.10	0.870

**Table 2 entropy-21-00401-t002:** PSNR and SSIM performances for the proposed denoising method in different standard images with σ={10,20,30,40,50} of Additive White Gaussian Noise (AWGN).

Image	Noise(*σ*)	Proposed Denoising Method													
		HAAR		DB2		DB4		SYM2		SYM4		COIF2		COIF4	
		PSNR	SSIM	PSNR	SSIM	PSNR	SSIM	PSNR	SSIM	PSNR	SSIM	PSNR	SSIM	PSNR	SSIM
**Lena**	10	34.16	0.881	34.22	0.883	**34.98**	**0.883**	34.14	0.878	33.92	0.874	34.15	0.880	34.08	0.878
	20	32.91	0.854	32.46	0.854	**33.02**	**0.857**	32.19	0.850	32.08	0.848	32.38	0.851	32.36	0.852
	30	29.90	0.816	30.00	0.819	**30.07**	**0.822**	29.91	0.812	29.86	0.812	29.89	0.814	29.88	0.813
	40	28.71	0.779	28.70	0.779	**28.78**	**0.788**	28.66	0.772	28.39	0.764	28.45	0.768	28.42	0.766
	50	27.36	**0.703**	27.40	0.699	**27.43**	0.701	27.34	0.689	27.29	0.686	27.31	0.686	27.33	0.684
Jetplane	10	35.05	0.919	35.14	**0.922**	**35.17**	**0.922**	35.01	0.906	34.89	0.902	34.92	0.910	34.99	0.904
	20	31.86	0.871	31.87	0.873	**31.92**	**0.874**	31.82	0.865	31.80	0.862	31.84	0.864	31.84	0.863
	30	29.92	0.819	29.92	0.821	**29.96**	**0.827**	29.90	0.816	29.87	0.815	29.91	0.817	29.90	0.815
	40	28.24	**0.788**	**28.25**	0.782	**28.25**	0.784	28.20	0.762	20.18	0.761	28.18	0.763	28.18	0.763
	50	27.09	**0.745**	27.08	0.744	**27.11**	0.744	27.03	0.726	26.98	0.724	27.00	0.724	27.01	0.726
Mandrill	10	34.39	0.922	34.44	0.922	**34.46**	**0.924**	34.36	0.913	34.35	0.905	34.37	0.917	34.36	0.909
	20	31.67	0.837	31.71	0.844	**31.75**	**0.847**	31.63	0.832	31.61	0.826	31.63	0.834	31.62	0.829
	30	29.56	0.779	29.57	0.786	**29.61**	**0.791**	29.54	0.763	29.53	0.763	29.54	0.771	29.56	0.773
	40	27.89	**0.705**	27.93	0.701	**27.94**	0.704	27.86	0.692	27.85	0.688	27.88	0.691	27.87	0.687
	50	27.08	**0.661**	27.11	**0.661**	**27.15**	0.660	27.05	0.650	27.01	0.639	27.03	0.654	27.03	0.638
House	10	39.21	0.939	**39.23**	0.944	**39.23**	**0.945**	39.16	0.936	39.15	0.921	39.17	0.937	39.18	0.924
	20	36.24	0.911	36.30	0.914	**36.33**	**0.919**	36.22	0.904	36.19	0.901	36.22	0.903	36.19	0.902
	30	34.66	0.867	34.67	0.863	**34.74**	0.866	34.55	**0.869**	34.52	0.865	34.59	0.867	34.58	0.866
	40	33.13	**0.851**	33.12	0.842	**33.17**	0.847	33.03	0.840	32.98	0.827	32.99	0.832	33.03	0.828
	50	31.67	**0.836**	31.70	0.834	**31.72**	**0.836**	31.64	0.819	31.64	0.814	31.65	0.818	31.67	0.818
Boat	10	34.00	**0.891**	34.17	0.883	**34.19**	0.889	33.94	0.869	33.91	0.864	33.93	0.871	33.94	0.866
	20	31.83	0.821	31.85	0.825	**31.88**	**0.829**	31.79	0.812	31.71	0.804	31.77	0.816	31.77	0.806
	30	29.67	0.766	**29.69**	**0.769**	**29.69**	0.767	29.49	0.758	29.48	0.755	29.53	0.759	29.58	0.759
	40	28.00	**0.723**	28.04	0.721	**28.07**	**0.723**	27.92	0.709	27.90	0.700	27.97	0.707	27.96	0.704
	50	26.77	0.651	26.77	0.647	**26.79**	0.648	26.70	0.638	26.69	0.629	26.72	0.636	26.76	0.633
Lake	10	33.07	0.888	33.07	**0.897**	**33.08**	**0.897**	32.96	0.876	32.95	0870	32.98	0.875	32.99	0.867
	20	30.29	0.823	30.33	**0.829**	**30.37**	0.826	30.26	0.819	30.26	0.809	30.26	0.817	30.28	0.811
	30	28.78	0.780	28.79	0.783	**28.81**	**0.784**	28.73	0.766	28.72	0.766	28.76	0.774	28.75	0.769
	40	27.34	**0.739**	27.39	0.737	**27.44**	**0.739**	27.30	0.724	27.28	0.723	27.31	0.720	27.33	0.721
	50	26.76	**0.701**	26.84	0.696	**26.88**	0.699	26.77	0.689	26.72	0.689	26.74	0.691	26.74	0.692
Peppers	10	34.79	0.879	34.84	**0.888**	**34.85**	0.882	34.77	0.868	34.76	0.856	34.77	0.863	34.78	0.863
	20	33.45	0.821	33.46	0.826	**33.49**	**0.827**	33.44	0.817	33.39	0.811	33.42	0.808	33.42	0.809
	30	31.37	0.781	31.36	0.781	**31.40**	**0.783**	31.33	0.779	31.28	0.769	31.34	0.772	31.33	0.770
	40	30.09	**0.746**	**30.10**	0.742	**30.10**	**0.746**	29.91	0.733	29.87	0.732	29.88	0.734	26.87	0.732
	50	28.87	**0.716**	28.90	0.713	**28.94**	0.715	28.85	0.698	28.82	0.681	28.83	0.697	28.81	0.694
Barbara	10	33.39	0.886	33.39	0.886	**33.41**	**0.888**	33.34	0.877	33.30	0.875	33.35	0.876	33.31	0.879
	20	29.02	0.829	29.07	0.829	**29.17**	**0.831**	29.00	0.817	28.97	0.812	28.98	0.813	29.01	0.815
	30	27.00	0.761	**27.02**	0.760	**27.02**	**0.763**	26.89	0.750	26.76	0.744	26.72	0.747	26.74	0.749
	40	25.61	**0.726**	25.61	0.724	**25.65**	0.725	25.54	0.719	25.52	0.712	25.52	0.714	25.50	0.714
	50	24.98	**0.678**	**25.02**	0.672	**25.02**	0.674	24.97	0.656	24.92	0.655	24.95	0.658	24.94	0.661
Pirate	10	34.10	0.887	34.12	**0.890**	**34.17**	**0.890**	34.06	0.876	34.00	0.876	34.05	0.879	34.06	0.875
	20	31.94	0.827	**32.01**	0.833	**32.01**	**0.835**	31.86	0.820	31.83	0.814	31.88	0.808	31.89	0.813
	30	29.46	0.779	29.51	**0.781**	**29.52**	0.780	29.44	0.755	29.41	0.752	29.44	0.756	29.42	0.752
	40	28.64	**0.722**	28.69	0.720	**28.72**	0.720	28.61	0.711	28.59	0.710	28.61	0.712	28.60	0.708
	50	26.78	**0.686**	**26.79**	0.677	**26.79**	0.682	26.73	0.659	26.71	0.662	26.73	0.663	26.72	0.664
Texture	10	32.35	0.936	32.48	0.943	**32.49**	**0.944**	32.36	0.928	32.35	0.924	32.33	0.928	32.34	0.928
	20	28.07	0.885	28.11	0.889	**28.14**	**0.893**	28.04	0.877	28.04	0.869	28.06	0.875	28.04	0.873
	30	26.28	0.823	26.33	0.822	**26.36**	**0.824**	26.27	0.807	26.25	0.800	26.27	0.803	26.26	0.807
	40	24.11	**0.737**	**24.15**	0.731	**24.15**	0.733	24.05	0.722	24.02	0.722	24.04	0.719	24.05	0.721
	50	22.88	**0.681**	22.97	0.674	**23.03**	0.676	22.88	0.663	22.83	0.660	22.87	0.664	22.86	0.661
Average	10	34.49	0.902	34.51	0.905	**34.60**	**0.907**	34.41	0.892	34.35	0.886	34.40	0.893	34.38	0.889
	20	31.72	0.847	31.71	0.851	**31.81**	**0.853**	31.62	0.841	31.58	0.835	31.64	0.838	31.64	0.837
	30	29.67	0.797	29.68	0.798	**29.71**	**0.800**	26.66	0.786	29.56	0.784	29.59	0.788	29.60	0.783
	40	28.28	**0.751**	29.09	0.749	**29.12**	0.750	28.10	0.738	27.25	0.733	28.08	0.736	27.78	0.734
	50	27.02	**0.705**	27.05	0.701	**27.08**	0.703	26.99	0.688	26.96	0.683	26.98	0.689	26.96	0.687

Note: The best results are given in bold format.

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
