# Peer review of "Local Complexity Estimation Based Filtering Method in Wavelet Domain for Magnetic Resonance Imaging Denoising"

_entropy, 2019, doi:10.3390/e21040401_

Round 1

Reviewer 1 Report

In this paper, the authors introduced the local complexity estimation-based filtering method in wavelet domain for MRI (Magnetic resonance imaging) denoising. A threshold selection methodology is proposed in which the edge and detail preservation properties for each pixel are determined by the local complexity of the input image.

The idea behind this is interesting. However, I still have quite a number of concerns in this manuscript. There are times where there are not enough data to support the conclusions of the author. Please see some of the major concerns below.

1.The information for the MRI data is not enough. The authors should give much more information about this. For examples did data was taken from an MRI machine? ,If yes which type? And how much images were taken? . Also are the images taken from the same patient or there are many patients? . So the readers can get its reproducibility. 

2.  The authors should give much more information about the novelty of this paper, especially the effect of using this new filtering method, which MRI applications can be used?

3. The authors claim that they done an experiment (section 3 Experimental Results), but they didn’t give any data (setup image or sketch of the AWGN with the MRI), I think they need to change this section to simulation results, which is more appropriate in this case.

4. For the readers, it will be good to include all statistical analysis in the paper.

5. More references need to be included in the introduction part to understand the applications of MRI algorithm and images detections.  

(a)"Improved diagnostic process of multiple sclerosis using automated detection and selection process in magnetic resonance imaging, " Applied Sciences, Issue 7(8), 831, (2017).

(b) Color image identification and reconstruction using artificial neural networks o

multi-mode fiber images: Towards an all- optical design

6.  Much more discussion about the results should be given in this paper, especially the author needs to provide enough physicals mechanism analysis about the results. Which mean are this new method can be helpful for diagnostic of any kind of disease?

Author Response

Thanks very much for your comments, please find the responses in the attachment.

Author Response

(The authors gave the same response as above.)

Reviewer 3 Report

This is a well-written paper. The paper proposed a filtering method based on local complexity estimation in the wavelet domain for MRI denoising. The method is well described with robust experiments support.  

However, the proposed method was compared with some old denoising methods (2005, 2008, 2012). There is a branch of new MRI denoising algorithms published recently (e.g., Bayesian MRI denoising in the complex domain, 2017; Denoising of 3D magnetic resonance images with multi-channel residual learning of convolutional neural network, 2018). The authors should compare with the most recent algorithms.

Other minor typos. E.g. page 2, second paragraph, line 3: a closing parenthesis missing; page 6, line 3: “let obtain …”; second paragraph, line 4: “let compute…”; last third line: “color retention” format; 

Author Response

(The authors gave the same response as above.)

Round 2

Reviewer 1 Report

The new version of the paper can be published 

Reviewer 2 Report

Authors addressed all the raised points in a satisfacotry way,